# Unusual Case of Masseter Muscle Hypertrophy in Adolescence—Case Report and Literature Overview

**DOI:** 10.3390/diagnostics12020505

**Published:** 2022-02-16

**Authors:** Elena Ţarcă, Elena Cojocaru, Alina Costina Luca, Laura Mihaela Trandafir, Solange Tamara Roşu, Valentin Munteanu, Viorel Țarcă, Cristian Constantin Budacu, Claudia Florida Costea

**Affiliations:** 1Department of Surgery II-Pediatric Surgery, “Grigore T. Popa” University of Medicine and Pharmacy, 700115 Iaşi, Romania; elena.tuluc@umfiasi.ro; 2Department of Morphofunctional Sciences I—Pathology, “Grigore T. Popa” University of Medicine and Pharmacy, 700115 Iaşi, Romania; 3Department of Mother and Child Medicine–Pediatrics, “Grigore T. Popa” University of Medicine and Pharmacy, 700115 Iaşi, Romania; alina.luca@umfiasi.ro (A.C.L.); laura.trandafir@umfiasi.ro (L.M.T.); 4Department of Nursing, “Grigore T. Popa” University of Medicine and Pharmacy, 700115 Iaşi, Romania; solange.rosu@umfiasi.ro; 5Department of Biomedical Sciences, “Grigore T. Popa” University of Medicine and Pharmacy, 700115 Iaşi, Romania; valentin_munteanu2002@yahoo.com; 6Department of Communication Sciences, Apollonia University, 700613 Iasi, Romania; vtarca@gmail.com; 7Department of Dentoalveolar and Maxillofacial Surgery, “Grigore T. Popa” University of Medicine and Pharmacy, 700115 Iaşi, Romania; 8Department of Surgery II-Ophthalmology, “Grigore T. Popa” University of Medicine and Pharmacy, 700115 Iaşi, Romania; claudia.costea@umfiasi.ro

**Keywords:** unilateral idiopathic masseter muscle hypertrophy, children, facial esthetics

## Abstract

Unilateral hypertrophy of the masseter muscle is a very rare pathological entity in children. Its etiology is uncertain and it requires a high degree of suspicion, as it must be differentiated from other conditions of the masseter area. As there are few pathological studies to elucidate this condition, we report a rare case of unilateral masseter muscle hypertrophy in a 16-year-old female patient with gradual onset of a painless swelling in the posterior left cheek which caused facial asymmetry with repercussions on the patient’s self-image. The diagnosis of unilateral masseter muscle hypertrophy was suggested by clinical examination, ultrasound scanning, and nuclear magnetic resonance, and was confirmed by histologic examination two years later when the patient returned for the surgical correction. The pathological findings report showed fragments of skeletal muscle with hypertrophic fibers associated with normal-sized muscle fibers in both longitudinal and transverse sections. The postoperative evaluation was favorable as both the adolescent and her family were satisfied with her look on the 14th day, 1st year, and 3rd year follow-ups. In conclusion, unilateral masseter muscle hypertrophy in adolescence is a sensitive problem due to the psychological implications of facial appearance. Definite diagnosis and treatment of the hypertrophied muscle is the ideal solution.

## 1. Introduction

Both women and men, especially adolescents, have always been influenced by certain cultural standards of beauty; e.g., an oval or heart-shaped face may be considered beautiful, while a square and especially asymmetrical one is seen as ‘ugly’ and may even cause mental disorders. There is a worldwide tendency for smooth facial lines and a slim jawline as desirable in females; a squared jaw is considered more of a major aesthetic issue among Asians, but also Caucasians [1,2]. The lower facial contour is determined by the bony structure of the mandible covered by soft tissue, i.e., skin, subcutaneous cell tissue, and masticatory muscles. Abnormalities of these components may cause facial asymmetries, problems with facial appearance, and mental impairment. Bilateral hypertrophy of the masseter muscles is a rare pathological entity and was first described in 1880 in a 10-year-old girl [3]. Unilateral hypertrophy is even rarer and sometimes raises problems of differential diagnosis. Very few cases of unilateral hypertrophy in children have been reported in the literature, and very few pathological studies to elucidate this condition and the etiology of muscle hypertrophy have been published so far. A 2006 study reported 130 cases of masseter muscle hypertrophy reported in the literature and another 2016 study reported 140 cases; a 2018 study reported 30 female patients, all adults [4,5,6]. The condition affects both sexes, with a slight male predominance, and is more common between the ages of 20 and 40 in the Asian race [4].

From an anatomical point of view, the masseter muscle is a quadrilateral muscle made up of two layers; the top of which is inserted on the lower and deep edge of the zygomatic arch and the bottom of which is inserted on the inferolateral edge of the mandible. The masseter muscle is the bulkiest and strongest muscle of the face and maximum force is applied to it during mastication. Muscle hypertrophy means an increase in the size of individual muscle fibers. Masticatory muscle hypertrophy is generally considered to be a functional hypertrophy and can affect all, several, or just one of the muscles of mastication; it can occur either bilaterally or unilaterally. In some cases, other masticatory muscles may be impaired, such as the temporalis or internal pterygoid muscle. Periosteal appositions, revealed by medical imaging, may occur on the lower branch of the mandible on the affected side due to excessive bone traction, periosteal irritation, and new bone deposition. However, approximately 20% of normal people have this finding and it cannot be considered as a diagnostic aid [7]. Patients rarely complain of localized pain in the affected area or dental mal occlusion; the most common complaints are of an aesthetic nature.

Although the etiology of this condition is still unclear, certain factors, such as temporomandibular joint disorders, stomatognathic system dysfunction, bruxism, mental disorders, or excessive use of chewing gum are thought to be involved; unilateral occurrence can be seen when patients chew or clench primarily on one side [8,9,10]. Bruxism may be a risk factor of malocclusion, generating higher forces due to the increased activity of the masticatory muscles; bruxism may also cause some alterations in the dimensions of the maxilla, leading to a larger palate, causing facial asymmetry and masticatory muscle hypertrophy [11]. Oral submucous fibrosis, a pre-malignant condition, also may manifest with masseter muscle hypertrophy [9]. According to Teixeira et al., there are two types of masseter muscle hypertrophy: congenital or familial and acquired due to functional hypertrophy [12]. Reactive hypertrophy develops when the masticatory muscle workload is increased by local bone and dental disorders; such triggers are not powerful but act over long periods, thus demanding increased endurance [13,14]. Zachariades et al. speculated that a vascular lesion may gradually subside to a residual muscular hypertrophy; they reported two cases in which phleboliths were associated with masseteric hypertrophy [15].

When the condition is bilateral or unilateral, its differential diagnosis will include diseases of the parotid glands (inflammation, tumors, infections, autoimmune diseases), dental or osteoarticular conditions, muscle tumors, lipomas, myopathy, vascular tumors, or genian hemangioma [16,17]. These conditions may be ruled out by biological tests and medical imaging scanning; in unilateral cases, the definite diagnosis is even more difficult and pathological tests are sometimes necessary.

There is not yet a standard protocol in the literature about the treatment of masseter hypertrophy. The treatment modalities for masseter hypertrophy can be nonsurgical and surgical. Management of the idiopathic masseter hypertrophy is based on psychological counseling, the use of mouth guards, muscle relaxant and anxiolytic drugs, analgesics, physical therapy, dental restorations, and occlusal adjustments to correct premature contacts [10]. When the condition is bilateral and symmetrical, reassuring patients of the benign nature of the hypertrophy or administering analgesics, muscle relaxants, or psychological counseling may be sufficient [17]. In some cases, mainly when stress is involved, psychological follow-up may be required in association with other treatments [18]. In addition to the problems with facial appearance and mental impairment, if left untreated, masseter muscle hypertrophy may cause other complications: Reddy et al. report three cases of obstructive parotitis secondary to an acute bend in the Stensen’s duct caused by masseter muscle hypertrophy [19].

However, conservative therapy is often not effective and invasive treatment is necessary to improve dental occlusion, especially in unilateral cases in adolescents, as was our case, when surgery was demanded for aesthetic reasons. The treatment consists of surgical procedures, namely the resection of excessive vertical fibers from the inner third of the hypertrophied muscle and mandibular osteotomy in cases of bony hyperplasia of the mandibular angle. Surgical treatment was first described by Gurney in 1947 [20,21]. The surgical approach may be intraoral, extraoral, or combined, depending on whether only the resection of the internal area of the hypertrophied muscle is necessary, or whether bone resection is also required. The choice between intra- and extraoral approaches is not related to the cosmetic or functional outcomes or to the risk of introducing vascular and nerve injury, but to the skill and experience of the surgeon in performing surgery using either of the approaches [10]. Removal of the masseter muscle insertion by means of a triangular incision was done by Martensson in a patient with a history of bruxism and unilateral masseter muscle hypertrophy [22]. In the case of an intraoral approach, an internal muscle band is removed from the hypertrophied masseter from an upper insertion in the zygomatic arch to a lower insertion in the mandibular arch, thus avoiding the production of a visible scar on the patient’s face and reducing the possibility of injuring branches of the facial nerve [23].

When surgery is not enough, botulinum toxin injections can be used as a supplement [24]. Since 1994, intramuscular injection of botulinum toxin type A has been used in some centers for treating bruxism and facial asymmetry, a method that prevents surgery complications, with generally positive results [25,26,27]. Botulinum toxin type A is a complex bacterial protein that blocks muscle neurotransmitters, causing functional denervation of the masseter muscle and its subsequent atrophy [26]. Botulinum toxin is applied to the chewing muscles in patients who experience chronic pain or in patients suffering from facial asymmetry, despite the conventional treatments, to prevent excessive contractions in the muscles; it produces localized paralysis by blocking the release of acetylcholine at the neuromuscular junction without producing undesirable systemic effects [27]. The drawbacks of this method are that it can cause limitations or asymmetry of the smile, may result in decreased muscle strength and chewing difficulties, or iatrogenic masseteric bulging, and also that it requires repeated injections every 4–8 months. Moreover, the condition may reoccur by hyperplasia of muscle fibers [28]. Ultrasonography can be used to observe the internal structure of the masseter muscle and will help to reduce the side effects of masseteric bulging when applying retrograde or dual-plane injection of botulinum toxin [2]. Unlike the surgical excision of muscle tissue that reduces the actual number of muscle cells, botulinum toxin type A only reduces muscle volume temporarily [4,28]. Another minimally invasive alternative is tissue coagulation with high radiofrequency energy, but there is not yet enough data on its long-term efficacy [29,30].

### Justification of the Case Report

Because unilateral masseter muscle hypertrophy is a very uncommon pathological condition in children, the differential diagnosis may be difficult and pathological investigations to elucidate this disorder are scarce. A rare case of unilateral masseter muscle hypertrophy in a pediatric patient is reported here.

## 2. Case Presentation

The case of a 16-year-old female patient who came to the specialized outpatient clinic of “St. Mary” Emergency Children’s Hospital of Iași, Romania on 22 July 2016 for the gradual onset, over a period of about three years, of a painless swelling in the posterior left cheek, in the masseter muscle area, which caused facial asymmetry, with repercussions on the patient’s self-image is reported. The patient was hospitalized in the Pediatric Surgery Department, for further tests. Her clinical examination revealed that her facial asymmetry became even more visible when she clenched her teeth, which was even more aesthetically unpleasing (Figure 1).

Anamnesis revealed that she chewed on both sides, did not generally use chewing gum that there was no history of face injury, dental conditions, mental disorders, or parafunctional activities such as bruxism. No local inflammatory phenomena were detected and the consistency of the area was normal, homogeneous, painless, and with no signs of abnormalities due to dental occlusion. The initial differential diagnosis included a condition of the salivary glands, a lymphatic or arteriovenous malformation, a muscle tumor, or lipoma. All these conditions were ruled out with the help of clinical multidisciplinary investigations (psychological, neurosurgical, dentoalveolar and maxillofacial surgery, and ophthalmological consultation), biological tests, and medical imaging scanning (the findings were within the reference range). The ultrasound scanning of the masseter area revealed hypertrophied muscle tissue measuring 13/48 mm. Nuclear magnetic resonance (NMR) suggested the diagnosis of hypertrophy of the left masseter muscle as it was double in size compared to the contralateral muscle; muscle morphology and signal strength were normal, with no significantly high contrast agent uptake on the impaired side compared to the normal side (Figure 2). No other intracranial abnormalities or bone damage in the mandible were identified.

Both the patient and her parents were informed about the benign nature of the condition and the patient was discharged with the recommendation to use only the right side for chewing. The recommendations were followed but without any benefit and the adolescent returned about two years later with the same facial asymmetry and asked for surgery to fix it. The patient was readmitted to the Pediatric Surgery Department. Surgery was performed under general anesthesia with an external approach through a 3 cm-wide incision at the level of the lower edge of the mandible. for a favorable cosmetic appearance and to reduce the possibility of injuring branches of the facial nerve; the antero-external portion of the hypertrophied masseter muscle was highlighted and resection of excessive vertical fibers from the inner third was done; hemostasis was performed and the procedure ended with an intradermal suture and steri-strip dressing.

### 2.1. Follow-Up

After surgery, the patient suffered from a local hematoma that resorbed after 10 days of topical treatment. Both the adolescent and her family were very satisfied with the clinical aspect on the 14th day, 1st year, and 3rd year follow-ups (Figure 3). No functional loss was reported and no recurrence occurred after three years of follow-up.

### 2.2. Histological Assessment

The specimens removed by incision biopsy were standardly processed for histopathological examination at the Pathology Laboratory. The fragments cut from paraffin block were stained with hematoxylin-eosin (HE), PAS (Periodic acid–Schiff), and trichromic stains. We used morphometry to measure the diameter of the muscular fibers in the masseter muscles. The normal arrangement of the fibers is usually irregular, with a wide range of diameters less than 20 μm. The histological examination of the surgical specimen in our case revealed striated skeletal muscle with well-preserved architecture, but no evidence of interstitial inflammation. Fibers with a normal diameter (Figure 4) were found in some areas, as well as others displaying a substantial enlargement of the muscle cells with a wide range of diameters greater than 20 μm (Figure 5a,b). Thus, the diagnosis of left masseter muscle hypertrophy was confirmed, according to the examination sheet no. 64085 dated 28 June 2018.

## 3. Sources of Information

Using the Medical Subject Headings MeSH term “masseter muscle hypertrophy”, “idiopathic”, “unilateral”, and “children”, we performed a PubMed literature search for randomized controlled trials (RCTs), systematic reviews, observational studies, series of cases, and case reports from the earliest possible date to January 2022. Articles, and their reference lists, published in English were analyzed for other relevant articles. Searching PubMed for the term “masseter muscle hypertrophy”, 118 articles were found, but when the term “children” was added, only 39 articles returned. We reviewed the title and the abstract of these articles and only 24 of them were identified as effectively analyzing masseter muscle hypertrophy cases including children, and met the inclusion criteria. When searching PubMed for the term “unilateral masseter muscle hypertrophy”, 49 articles were found. Searching for “unilateral” and “idiopathic masseter muscle hypertrophy” yielded 19 articles. Searching for the term “masseter muscle hypertrophy” on EMBASE, title or abstract, from the earliest possible date to January 2022, returned 87 articles, but when we added the term “children”, only 3 articles returned. Searching for “unilateral masseter muscle hypertrophy” resulted in 2 articles. We checked the full text of all these articles and extracted data on incidence, sex ratio, diagnosis, etiology, associated abnormalities, and treatment. Thirty-eight of all the articles found were quoted.

## 4. Discussion

Unilateral masseter muscle hypertrophy is a very rare pathological entity in children. Its etiology is uncertain and it requires a high degree of suspicion, as it must be differentiated from other conditions of the masseter area. Medical imaging may help set the diagnosis, but pathological tests provide important details and help find the etiological cause. When treatment is planned, the patient’s expectations and physical findings should be evaluated thoroughly.

The unusual case of a teenage girl with unilateral masseter muscle hypertrophy, without other associated conditions, who, for aesthetic and psychological reasons, demanded surgery after two years of conservative treatment without benefits, is reported here. Her postoperative evolution was positive at the 3rd year follow-up. The condition is a rare pathological entity, being rather bilateral than unilateral, and its occurrence in adolescence is unusual. A study by Riefkohl et al. showed that of the total 90 patients, 4% were less than 10 years old, 3% were over 40 and the remaining patients had a mean age of 30 years [31]. There are very few cases of unilateral idiopathic masseter muscle hypertrophy in children and, as far as we know, this is the first case presented in detail including investigations, treatment, postoperative evolution, and pathological examination. In 2019, Antunes et al. presented a similar case on a female adolescent 15-year old, but the treatment, evolution, or pathological examination were not specified in the report [32]. Another case report is from 2016, a 17-year old female adolescent, but she did not receive any medical or surgical treatment [5]. Trujillo et al. reported a similar case in 2002 of a 16-year old boy; the chief concern of his grandmother was that the swelling could be a malignant tumor. The esthetic factor was not as important to them, so the patient was referred for relaxation therapy and no surgical intervention was indicated [18]; the authors do not mention how the patient evolved, there is no follow-up.

There are other cases presented in adults, but once more, no pathological examination of hypertrophied masseter muscle was found. In the presented case, none of the known etiological factors were identified; therefore, the condition was classified as idiopathic.

The diagnosis may be clinical, by palpating the masseter muscle with fingers while the patient clenches his/her teeth so the muscle is more prominent during contraction [10]. The masseter muscles are most typically injured, they are variably enlarged and they can be up to three times the size of the contralateral muscles. At imaging, they are normal in appearance without increased contrast [13]; periosteal appositions may occur and be visible on the lower branch of the mandible on the affected side, yet these were not detected in the case reported here. Early diagnosis of masseteric hypertrophy is important so that the patient and parents can be informed about the likely development of facial asymmetry.

The positive diagnosis of hypertrophy is established by the histological examination of the affected muscle [14]. In 2014, Katsetos et al. stated that ”Muscle biopsy is key in the diagnosis of reactive masticatory muscle hypertrophy and its distinction from masticatory muscle myopathy (hypertrophic branchial myopathy) and other non-reactive causes of painful asymmetric temporalis muscle enlargement” [33]; he and his team performed a thorough pathological examination of the affected temporalis muscle, with histochemical documentation of muscle hypertrophy with fiber type 1 predominance, which was difficult to establish in our case as we did not have the opportunity to perform histochemical staining to differentiate them. Still, some patients are treated symptomatically and a muscle biopsy is not performed due to various reasons [34,35]. On histological evaluation, the benign nature of the condition was demonstrated for the presented case; the serial histological sections showed hypertrophied striated skeletal muscle with well-preserved architecture, but no evidence of interstitial inflammation. In previous reports, histological examination of specimens from patients with masseter muscle hypertrophy demonstrated substantial hypertrophy with a muscle fiber diameter increase more than two to three times that of normal muscle fibers [36,37]. These results are in line with what we found in our case report but in slight contradiction with the Trujillo et al. statement that “microscopic examination of the removed muscle tissue usually shows normal muscle fibers without changes in length, thickness, or nuclear structure” [18].

Other conditions were ruled out in our case by biological tests and medical imaging scanning, but the definite diagnosis was set by anatomopathological means. In unilateral cases, the definite diagnosis is even more difficult, which makes pathological tests mandatory.

Since there is no standard protocol in the literature for the treatment of masseter muscle hypertrophy, our patient was treated conservatively for two years with no benefit. Surgery was then performed using an extraoral approach with intradermal sutures for a better cosmetic appearance. Surgery may pose some risks such as facial or mandibular nerve injuries, masseter artery injury and bleeding, bone lesions, infections, postoperative trismus, mouth opening limitation, and sequelae from general anesthesia [25,38]. Our patient developed a postoperative hematoma, which was resorbed after 10 days of conservative treatment, and the long-term follow-up showed favorable results. Careful follow-up is required because this condition can be recurrent, but no case of relapse after surgical treatment has been reported in the literature (which is not the case with botulinum toxin treatment).

## 5. Conclusions

Unilateral masseter muscle hypertrophy is a very rare pathological entity in children; the treatment can be conservative, but it is difficult to apply to adolescents due to the psychological implications of facial aesthetics; partial excision of the hypertrophied muscle or botulinum toxin injection may be the solution.

## Figures and Tables

**Figure 1 diagnostics-12-00505-f001:**
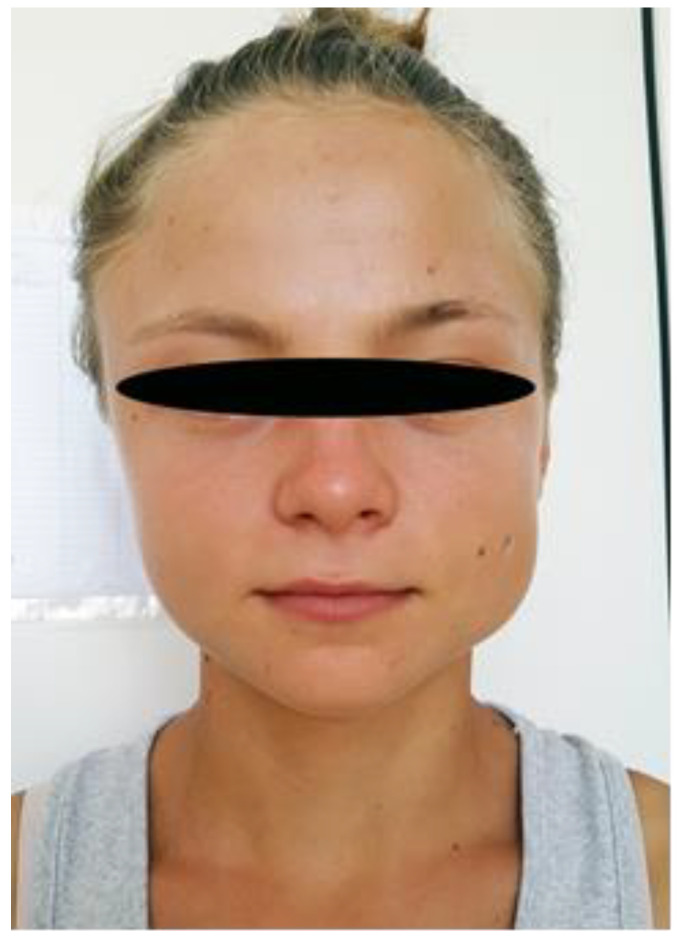
Hypertrophy of the left masseter muscle.

**Figure 2 diagnostics-12-00505-f002:**
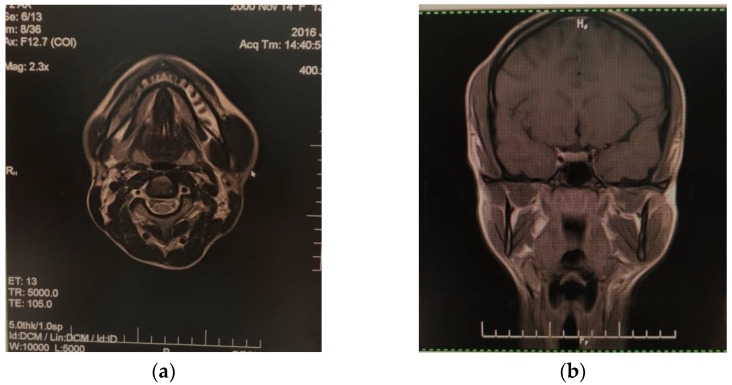
Nuclear magnetic resonance aspect of left masseter hypertrophy: (**a**) cross-plane; (**b**) coronal plane.

**Figure 3 diagnostics-12-00505-f003:**
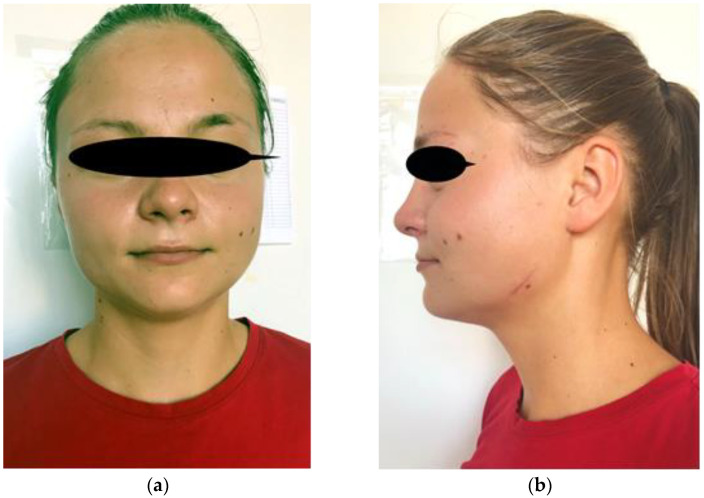
Postoperative appearance: (**a**) front; (**b**) profile.

**Figure 4 diagnostics-12-00505-f004:**
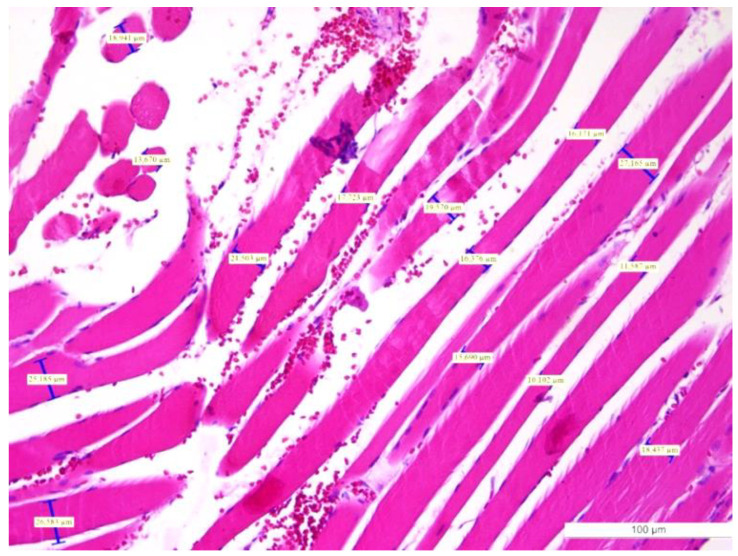
The histopathologic aspect of masseter muscle hypertrophy ((**left**) part of the image) and normal fibers ((**right**) part of the image). (Scale bars: 100 μm) Normal striated fibers—longitudinal section (HE staining, ×10). HE: Hematoxylin–Eosin.

**Figure 5 diagnostics-12-00505-f005:**
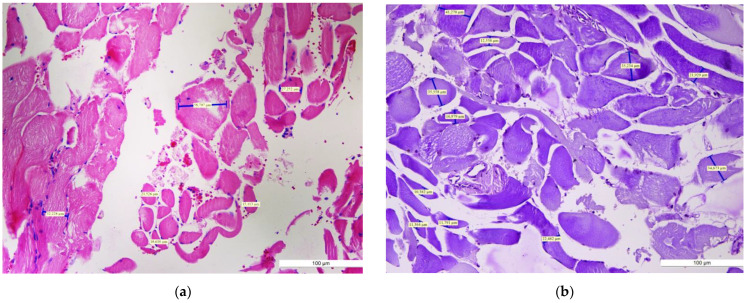
Histopathologic examination of masseter muscle hypertrophy: irregular arrangement of muscle fibers with sizes >20 µm (Scale bars: 100μm). (**a**) (HE staining, ×10). HE: Hematoxylin–Eosin. (**b**) (PAS staining, ×10). PAS: Periodic Acid Schiff.

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
