# Peer review of "Unusual Case of Masseter Muscle Hypertrophy in Adolescence—Case Report and Literature Overview"

_diagnostics, 2022, doi:10.3390/diagnostics12020505_

Round 1
Reviewer 1 Report
This is a revised manuscript about masseter muscle hypertrophy. The case is quite rare and it is better to publish. The revised manuscript was greatly improved.
Author Response
Dear Reviewer,
Thank you for evaluating our manuscript and for your appreciation. This is a revised manuscript indeed.
Reviewer 2 Report
The authors reported an interesting case of unilateral masseter muscle hypertrophy.
The English language is fine. The case presentation is well structured.
However, some aspects need to be revised, as listed below:
- The review process is too scant by far. It actually represents a formal literature screening more than a literature review. Thus, I strongly suggest to perform a better structured review or to remove it completely, evaluating the possibility to publish a case report, according to the journal policy.
- The figure 2 should be replaced with a better one, avoiding to take a photo of the computer screen instead of acquiring a high resolution screenshot. I also suggest to add an image of the coronal plane at least.
- Figure 3 should be moved to page 6
- Figures 4-10: all figures have a low resolution. Please provide better images both along the manuscript and in separate files. Remove the unnecessary images and group the remaining together providing a comprehensive explanation of the main findings under each image.
- Delete the space between lines 252 and 255
Author Response
“The authors reported an interesting case of unilateral masseter muscle hypertrophy. The English language is fine. The case presentation is well structured. However, some aspects need to be revised, as listed below.”
Dear Reviewer, thank you for evaluating our manuscript. Your recommendations and comments help us improve our manuscript.
Responses
- “The review process is too scant by far. It actually represents a formal literature screening more than a literature review. Thus, I strongly suggest to perform a better structured review or to remove it completely, evaluating the possibility to publish a case report, according to the journal policy.”
- For improving our review, we searched for similar articles on EMBASE, and also extended the screened period until January 2022 (see Sources of information section). In this way we managed to find more articles and analyzed them (see Introduction and Discussion section), and we added 6 more references.
- “The figure 2 should be replaced with a better one, avoiding to take a photo of the computer screen instead of acquiring a high resolution screenshot. I also suggest to add an image of the coronal plane at least.”
- We replaced the figure 2 and added the coronal plane image
- “Figure 3 should be moved to page 6.”
- We did that
- “Figures 4-10: all figures have a low resolution. Please provide better images both along the manuscript and in separate files. Remove the unnecessary images and group the remaining together providing a comprehensive explanation of the main findings under each image.”
- We replaced the images with better ones, and grouped them as you suggested.
- “Delete the space between lines 252 and 255.”
- We did that, thank you.
Reviewer 3 Report
The x-ray data used in this case report are shown only as MRI data, and the following test results for additional diagnosis and screening are shown.
1. PRT/CT
2. Panoramic X-ray
3. Intraoral photos to check for caries
4. Blood test
As a result, it is judged that this study performed only diagnosis based on MRI, and there was no attempt to diagnose from various viewpoints.
Therefore, it appears that it is not suitable for submission to this journal.
Author Response
Dear Reviewer, thank you for evaluating our manuscript.
As we stated in our manuscript, diagnosis of masseter muscle hypertrophy was set on clinical examination, anamnestic, multidisciplinary consults, biological investigations, imagistic methods, and confirmed on pathological examination.
-“Her clinical examination revealed that her facial asymmetry became even more visible when she clenched her teeth, which was even more aesthetically unpleasing (Figure 1).” “No local inflammatory phenomena were detected, the consistency of the area was normal, homogeneous, painless, and with no signs of abnormalities due to dental occlusion.”
- „Anamnesis revealed that she chewed on both sides, that she does not generally use chewing gum, that there was no history of face injury, dental conditions, mental disorders or parafunctional activities such as bruxism.”
- „The initial differential diagnosis included a condition of the salivary glands, a lymphatic or arteriovenous malformation, a muscle tumor, or a lipoma. All these conditions were ruled out with the help of clinical multidisciplinary investigations (psychological, neurosurgical, dentoalveolar and maxillofacial surgery, ophthalmological consultation).”
- „Biological tests and medical imaging scanning (the findings were within the reference range). The ultrasound scanning of the masseter area revealed hypertrophied muscle tissue measuring 13/48 mm.”
- „Nuclear magnetic resonance (NMR) suggests the diagnosis of hypertrophy of the left masseter muscle, as it was double in size compared to the contralateral muscle; muscle morphology and signal strength are normal, with no significantly high contrast agent uptake on the impaired side compared to the normal side (Figure 2). No other intracranial abnormalities or bone damage in the mandible were identified.”
-“The specimens removed by incision biopsy were standardly processed for histo-pathological exam…. Thus, the diagnosis of left masseter muscle hypertrophy was confirmed.”
We did not felt necessary to perform a highly irradiating CT scan because we obtained all the necessary information other ways. The patient was checked for caries by a dentoalveolar and maxillofacial surgeon and was clean, so no intraoral photos were taken. Biological tests were within the normal range.
We tried to improve all the sections of our manuscript, so we will be grateful if you revised it again. Thank you very much for your time.
Reviewer 4 Report
I congratulate the author team on the successful management of the patient problem. the case report has the potential to be published in the journal.
- Instead of aim, add justification of the case report, concise the aim part, it is too lengthy.
- Was there any functional loss in the patient also.
- the asymmetry was evaluated for bony growth or not?
- The labels are invisible, within the figure.
- Enhance the narrative of the manuscript.
- whether the consent for treatment or publication from the patient was sought out or not?
- Reduce the conclusion to include and focus the main achievements of the study.
Author Response
“I congratulate the author team on the successful management of the patient problem. The case report has the potential to be published in the journal.”
Dear Reviewer, thank you for evaluating our manuscript. Your recommendations and comments help us improve our manuscript.
Responses
- Instead of aim, add justification of the case report, concise the aim part, it is too lengthy.
-We change the chapter 1.1 into Justification of the case report, and we concise it.
- Was there any functional loss in the patient also.
- No functional loss was reported and no recurrence occurred after three years of follow-up. We added this statement in the text (Follow up chapter).
- The asymmetry was evaluated for bony growth or not?
- We did not perform a CT scan, but no bone damages in the mandible were identified on NMR. We specified this in the manuscript.
- The labels are invisible, within the figure.
- We replaced the images with better ones, and grouped them as another reviewer suggested.
- Enhance the narrative of the manuscript.
- For improving our review, we searched for similar articles on EMBASE, and also extended the screened period until January 2022 (see Sources of information section). In this way we managed to find more articles and analyzed them (see Introduction and Discussion section), and we added 6 more references.
- Whether the consent for treatment or publication from the patient was sought out or not?
- Yes, written informed consent from the patient and the patient’s parents for the treatment and for publication of this report and accompanying images was obtained. This is specified in the manuscript.
- Reduce the conclusion to include and focus the main achievements of the study.
- We have applied your recommendations, thank you.
Round 2
Reviewer 2 Report
The authors followed almost all the submitted suggestions. However, the issue regarding the literature review is still present. The submitted manuscript actually represents a well presented case report with a literature overview. Thus, if the authors are not interested to further improve the review process, and if this suggestion meet the journal policy, I suggest to change the article title replacing "literature review" with "literature overview" in order to avoid any misleading information to the readers.
Author Response
Dear Reviewer, thank you for reevaluating our manuscript. We changed the article title replacing "literature review" with "literature overview", as you suggested.
Reviewer 3 Report
As an Origianl article, there are many shortcomings, but I agree with the other reviewer's opinion of accepting it because it is an article worth presenting as a case report.
Author Response
Dear Reviewer, thank you for reevaluating our manuscript. We are presenting our article as a case report.
This manuscript is a resubmission of an earlier submission. The following is a list of the peer review reports and author responses from that submission.
Round 1
Reviewer 1 Report
This article described with a case of masseter muscle hypertrophy in a young woman. It is an unusual case and deserves to be published as a case report.
However, the image in Figure 2 is an MRI image and not a CT image. This should be corrected. Also, I think you should elaborate more on the surgical technique in line 99 for the future surgeons.
Author Response
Dear Reviewer,
Here we provide the requested corrections and address your comments. The changes we have made in the manuscript are highlighted in red.
- „This article described with a case of masseter muscle hypertrophy in a young woman. It is an unusual case and deserves to be published as a case report.”
- We changed the manuscript type into case report.
- „However, the image in Figure 2 is an MRI image and not a CT image. This should be corrected.”
- We corrected
- „Also, I think you should elaborate more on the surgical technique in line 99 for the future surgeons.”
- This was done
Thank you very much for evaluating our manuscript.
Reviewer 2 Report
The paper reports a case of unilateral masseter muscle hypertrophy in an adolescent girl. The manuscript is in general good and can be improved :
- Present it only as a case report and not as a simultaneous literature review. For writing the introduction and discussion for any case report, performing a literature review is a part of it. Focus should remain on case report.
- Discussion is very comprehensively written. You should consider moving some text from discussion to introduction. At least 70 percent of the references should first be cited in the introduction and then, during your case discussion, you may again refer to them.
- Discussion should be focused on how your case demographics, presentation, diagnosis, treatment and follow up was similar or different as compared to previously reported cases.
- Avoid Syntax error. Too much of 'we' has been used.
- In the case presentation, what is M.C.M (We report the case of M.C.M)?
- Recheck references (such as 22,24 etc). Write journal names in abbreviated form.
Author Response
Dear Reviewer, thank you for evaluating our manuscript. Your recommendations and comments helped us improve our manuscript.
- “Present it only as a case report and not as a simultaneous literature review. For writing the introduction and discussion for any case report, performing a literature review is a part of it. Focus should remain on case report.”
- We changed the manuscript type into case report.
- “Discussion is very comprehensively written. You should consider moving some text from discussion to introduction. At least 70 percent of the references should first be cited in the introduction and then, during your case discussion, you may again refer to them.”
- A big part of the text was moved from the discussion section to introduction. The discussion section was rewritten as you recommended.
3. „Discussion should be focused on how your case demographics, presentation, diagnosis, treatment and follow up was similar or different as compared to previously reported cases.”
- The discussion section was rewritten as you recommended.
4. „Avoid Syntax error. Too much of 'we' has been used.”
- This was corrected
5. „In the case presentation, what is M.C.M (We report the case of M.C.M)?”
-M.C.M. are the initials of the patient’s name, but we remove this from the text
6. „Recheck references (such as 22,24 etc). Write journal names in abbreviated form.”
-The references were corrected